# The Next Frontier in Sarcoma: Molecular Pathways and Associated Targeted Therapies

**DOI:** 10.3390/cancers15061692

**Published:** 2023-03-09

**Authors:** Ted Kim, Nam Q. Bui

**Affiliations:** Division of Oncology, Department of Medicine, Stanford University, Stanford, CA 94305, USA

**Keywords:** sarcoma, cell cycle, angiogenesis, targeted therapies, immunotherapy

## Abstract

**Simple Summary:**

Soft tissue sarcomas are an extremely rare group of cancers from mesenchymal origin. STS are difficult to treat due to their large variety of histological subtypes that dictate both their individual tumorigenesis and clinical characteristics. Although traditional chemotherapy has remained the mainstay treatment for advanced and metastatic disease, targeted therapies have emerged as a promising new approach to treat STS based on the specific molecular pathways of the tumor. Further elucidating the molecular pathways implicated in the development of STS will help guide the development of promising new therapeutics that can effectively target these pathways. Numerous targeted therapies against STS are therefore being tested in clinical studies to evaluate the efficacy and safety of these new treatments.

**Abstract:**

Soft tissue sarcomas (STS) are a rare, complex, heterogeneous group of mesenchymal neoplasms with over 150 different histological subtypes. Treatments for this malignancy have been especially challenging due to the heterogeneity of the disease and the modest efficacy of conventional chemotherapy. The next frontier lies in discerning the molecular pathways in which these mesenchymal neoplasms arise, metastasize, and develop drug-resistance, thereby helping guide new therapeutic targets for the treatment of STS. This comprehensive review will discuss the current understanding of tumorigenesis of specific STS subtypes, including oncogenic pathway alterations involved in cell cycle regulation, angiogenesis, NOTCH signaling, and aberrant genetic rearrangements. It will then review current therapies that have been recently developed to target these pathways, including a review of ongoing clinical studies for targeted sarcoma treatment, as well as discuss new potential avenues for therapies against known molecular pathways of sarcomagenesis.

## 1. Introduction

Soft tissue sarcomas (STS) are rare and heterogenous malignancies of mesenchymal origin accounting for less than 1% of all adult solid malignancies [1]. They can arise anywhere in the body, with most originating in the extremities (59%), trunk (19%), and retroperitoneum (15%). Of the numerous different histological subtypes, the most common are undifferentiated pleomorphic sarcoma (formerly malignant fibrous histiocytoma) (28%), liposarcoma (15%), leiomyosarcoma (12%), and synovial sarcoma (10%). The histological grade of STS assigned according to the FNCLCC histological grading system remains the most important prognostic factor and is best determined by an experienced sarcoma pathologist. Higher grade STS can have a metastatic potential as high as 60% [2].

Surgical resection with quality surgical margins (en-bloc R0 resection) remains the mainstay in treatment for localized, resectable disease, with the possible addition of radiotherapy in the neoadjuvant or adjuvant setting to improve local control rates for large, high grade (Grade 2/3) tumors [3]. However, for patients with recurrent advanced or metastatic disease, conventional chemotherapy has been shown to demonstrate only modest activity, with generally low response rates and short progression-free survival (PFS) and overall survival (OS) times. In cases of patients with metastatic disease, the overall survival (OS) ranges between 12 and 19 months, with little improvement in OS over the last 20 years [4]. ANNOUNCE, a recent phase 3, double-blind, randomized trial found an OS of only ~20 months for patients with unresectable locally advanced or metastatic STS treated with conventional chemotherapy (NCT02451943) [5]. The advent of personalized and precision medicine has now focused efforts on the research and development of molecular targeted therapies aimed at targeting known molecular pathways of STS. New therapies developed over the past decade, such as Pazopanib, a multitargeted tyrosine kinase inhibitor, as well as other targeted therapies provide a promising new avenue for the treatment and management of STS [4,6,7].

The histological complexity of the STS and the associated varying degrees of prognostics further highlight the importance of developing more specific, targeted treatments for different histological subtypes of STS. It is therefore important to elucidate the various molecular pathways involved in sarcomagenesis to help further advance treatment options for this rare and complex disease.

## 2. Molecular Mechanisms and Targets for STS

### 2.1. Cell Cycle Regulation

#### 2.1.1. Basic Science

One of the most well recognized hallmarks of tumorgenesis is the rapid and uncontrolled growth of cancer cells. The dysregulation of cyclin-dependent kinase (CDK) signaling pathways is thought to be a strong driver of sarcomagenesis [8]. Cyclin-dependent kinases are serine/threonine kinases that play vital roles in the control and modulation of cell division and transcription in response to various extracellular and intracellular cues. CDKs require and depend on a regulatory subunit—a cyclin—that provides additional sequence domains essential for their enzymatic activity [9]. Humans have 20 CDKs and 29 different cyclins with which they interact. CDK1, CDK2, CDK3, CDK4, and CDK6 are part of a group that is primarily responsible for cell cycle progression, while CDK8, CDK9, CDK10, and CDK11 primarily mediate gene transcription. CDK7 contributes to both processes, while other CDKs that do not fit into these two groups are associated with diverse functions that are often tissue specific [10,11].

CDKs associated with cell cycle regulation orchestrate the transition between the four distinct phases of the mammalian cell cycle: G_1_ (cells prepare for DNA replication or cellular quiescence), S (DNA synthesis), G_2_ (cells prepare for mitosis), and M (mitosis). When activated by their regulatory subunit (cyclin), the activated cyclin–CDK complex phosphorylates substrates that drive the cell past checkpoint inhibitors to the next stage of the cell cycle [12]. Cell cycle checkpoints exist between each phase of the cell cycle, and their complexity contributes to their ability to adequately control vital attributes of cellular division, including controlling cell size, DNA repair, and DNA replication [13]. Humans have two distinct genetic families of CDK inhibitors (CDKIs) that regulate the cyclin–CDK complex, thereby coordinating the entry of the cell into different phases of the cell cycle. The INK4 gene family encodes the proteins p16^INK4a^, p15^INK4b^, p18^INK4c^, and p19^INK4d^,which bind CDK4 and CDK6, inhibiting their ability to interact with D-type cyclins required for the cell to move past the G_1_ checkpoint and into the S-phase of DNA replication. The second family consists of the Cip/Kip proteins, including p21^Cip1/Waf1/Sdi1^, p27^Kip1^, and p57^Kip2^, which are also able to mediate the association between cyclins and CDK proteins but have distinct portions of their sequence, which suggests cellular functions beyond cell cycle regulation [14,15].

#### 2.1.2. Treatment Applications

Dysregulation in CDK pathways has been closely characterized in several soft tissue sarcoma subtypes, including liposarcoma, leiomyosarcoma, undifferentiated pleomorphic sarcoma, and others. In fact, liposarcomas are defined by amplification of MDM2, which is frequently co-amplified with its genomic neighbor CDK4/6 [16]. When CDK4/6 is inhibited from associating with type D-cyclins, retinoblastoma (RB) protein is unable to be phosphorylated. RB is a tumor suppressor protein when in its hypophosphorylated state; it actively suppresses G_1_-S progression through sequestering E2F transcription factors needed by genes responsible for DNA replication. Loss of the p16^INK4a^ CDKI leads to the unregulated phosphorylation of RB by CDK4/6, thereby leading to unregulated G_1_-S progression of cellular division [11,12]. Inhibition of CDK4/6 is therefore a potential target for new therapeutic agents aimed at cell cycle inhibition. Since liposarcomas have very high frequencies of CDK4/6 amplification, there have been numerous clinical efforts and trials with CDK4/6 inhibitors. Palbociclib is a CDK4/6 inhibitor that has shown clinical success for the treatment of both well-differentiated and dedifferentiated liposarcoma in two separate clinical trials (Table 1). Patients treated with a 200 mg daily dose of palbociclib for days 1–14 of a 21-day cycle achieved a median PFS of 18 weeks and a 12-week PFS rate of 66% [17]. The second trial dosed patients on a lesser, 125 mg daily dose regimen of palbociclib for days 1–14 of a 21-day cycle, and the results from this phase II trial demonstrated a median PFS of 17.9 weeks and a 12-week PFS rate of 57.2% [18]. Supported by the results of these two clinical studies, palbociclib is currently a category 2a NCCN recommendation for the treatment of well-differentiated and dedifferentiated liposarcoma [16]. There are now new, more potent CD4/6 inhibitors that are being evaluated in ongoing randomized clinical trials, such as abemaciclib (NCT04967521) (Table 1). 

Additionally, dedifferentiated liposarcoma has been uniquely characterized by an amplification of *MDM2*. The MDM2 protein is associated with increased cell-cycle progression and malignant proliferation due to its role in targeting downstream tumor suppressor p53 for degradation [19]. Therefore, emerging therapeutics aimed at targeting the overexpression of the MDM2 protein are also being studied. Milademetan, a selective small-molecule inhibitor of the MDM2–p53 interaction, has been tested in a first-in-human phase I study in a subset of patients with dedifferentiated liposarocomas (NCT01877382) (Table 1). In this study, 22 of the 53 patients with dedifferentiated liposarcoma were confirmed to have *MDM2* amplification or overexpression. Treated patients with dedifferentiated liposarcoma experienced a median PFS of 7.2 months as compared to only 3.4 months in nonliopsarcoma tumors [20]. MANTRA, a randomized, multicenter, phase 3 trial of milademetan versus trabectedin is currently being studied in patients with dedifferentiated liposarcomas to evaluate the efficacy and safety of milademetan as compared to conventional chemotherapy (NCT04979442) [21]. 

### 2.2. Angiogenesis

#### 2.2.1. Basic Science

Angiogenesis is a process in which new blood vessels are formed from existing vessels in a normal adult. Typically, this complex multi-step process is in a quiescent state. In certain pathologies, however, the quiescent vasculature can become activated to grow new capillaries. Basement membranes surrounding the endothelial vessel tube are degraded, allowing local endothelial cells to invade the surrounding stroma, proliferate, and tightly adhere to each other, thereby forming a lumen of a new capillary vessel [22]. In the absence of adequate vasculature, tumor cells are unable to proliferate and increase in tumor volume, eventually becoming necrotic. 

The role of angiogenesis and its importance in the growth and metastatic nature of solid tumors have been well documented. A study published in the 1970s demonstrated that when small fragments of avascular tumor were implanted directly on the highly vascular iris of a rabbit eye, the tumor always vascularized in a phase-like manner, resulting in exponential malignant growth. Conversely, when similar implants were placed in the anterior chamber of the eye where capillaries were physically absent, the tumor implant did not vascularize and remained dormant at a small size. When reimplanted on the iris, the tumor implant vascularized, resulting in rapid growth, suggesting that angiogenesis may be a necessary condition in which solid tumors are able to proliferate [23]. 

#### 2.2.2. Treatment Applications

Subsequent studies have continued to support the notion that angiogenesis may play a vital role in tumorigenesis, motivating research towards the search for potential inducers of angiogenesis and their correlation with clinical features of STS. High expression of tumor vascular endothelial growth factor (VEGF) is one that has been intricately correlated with higher tumor grade [24]. More specifically, evaluation of VEGF expression in gastrointestinal stromal tumor (GIST) using immunohistochemistry revealed that increased VEGF tumor expression was associated with inferior outcomes such as increased risk of liver metastases [25]. Additionally, antiangiogenic therapies have shown promising antitumor activity in animal models of sarcomas [26]. These results may suggest that VEGF expression may not only drive tumor vessel growth but may facilitate tumor metastasis by increasing vascular permeability [27]. Anti-angiogenic therapeutics therefore are now used and studied for their efficacy in certain STS histological subtypes. Pazopanib is the first and only tyrosine kinase inhibitor that is currently approved for the treatment of multiple histological subtypes of STS (Table 1). The landmark phase 3 randomized study, PALETTE, demonstrated improved progression-free survival with pazopanib for metastatic non-adipocytic STS after previous chemotherapy, leading to its approval [6]. In clinical practice, we have noticed patients who progress on chemotherapy and have a good response to pazopanib, suggesting that switching the mechanism of action is useful in STS treatment. In addition, the radiographic response patterns of VEGFi, such as pazopanib, differ from chemotherapy in that lesions can become cystic first, before shrinking. Pazopanib has also demonstrated significant clinical activity in desmoid tumors, with improvements in clinical symptoms such as pain, with a manageable toxicity profile [28]. 

Other antiangiogenic therapeutics have been thoroughly studied in various tumor types as well. For instance, regorafenib, an orally bioavailable multikinase inhibitor that also targets VEGFR-1, VEGFR-2, and VEGFR-3 has shown promising evidence of activity in the treatment of metastatic osteosarcoma in adult patients who have failed previous chemotherapy (NCT02389244) (Table 1) [29]. Cabozantinib, an inhibitor of MET and VEGFR-2, has been studied in Ewing sarcoma and osteosarcoma in the CABONE phase 2 study, where it demonstrated antitumor activity in both tumor groups (NCT02243605) (Table 1) [30]. Sorafenib, another multikinase inhibitor, has demonstrated clinical utility in patients with desmoid tumor, with an objective response rate of 33% and significant improvements in quality of life (NCT02066181) (Table 1) [31].

### 2.3. Growth Factors

#### 2.3.1. Basic Science

Constitutive activation of growth factors, including platelet-derived growth factor (PDGF), insulin-like growth factor (IGF), fibroblast growth factor (FGF), and epidermal growth factor (EGF), result in the activation of the downstream PI3K/PTEN/AKT/mTOR pathway. The dysregulation of the mammalian target of the rapamycin (mTOR) pathway and its role in sarcomagenesis has been characterized. mTOR is involved in translating proteins required for cell-cycle progression, growth, and survival [12]. mTOR is part of the serine-threonine protein PI3K-related kinases found in two distinct multiprotein complexes, mTORC1 and mTORC2 [32]. 4E-binding protein (4EBP) and S6K are both known to be downstream targets of mTORC1 heavily involved with translational initiation and control, resulting in increased cell mass and autophagy via increased protein, ribosome, and lipid synthesis. mTORC2 downstream targets are less elucidated; however, they are thought to be important in the activation of AKT and the protein kinase C (PKC) family. mTOR and its upstream regulators have therefore emerged as an exciting new therapeutic target for the treatment of STS [33]. 

#### 2.3.2. Treatment Applications

##### *nab*-Sirolimus for Malignant Perivascular Epithelioid Cell Tumors

PEComas are a rare family of mesenchymal neoplasms, including angiomyolipoma (AML) and lymphangiomyomatosis (LAM), that are morphologically and immunophenotypically similar in nature [34]. They are defined by the presence of perivascular epithelioid cells co-expressing both muscle and melanocytic markers and are more commonly found in females, although much of the data is limited to case studies due to their extreme rarity. A subset of PEComas can exhibit malignant behavior, with the presence of locally invasive recurrences or the development of metastatic disease, most commonly in the lung. Previously, no effective therapies for malignant PEComas have been described. However, tumors of the PEComa family have been observed at a higher frequency in patients with tuberous sclerosis complex (TSC) caused by mutations of *TSC1* or *TSC2*, which negatively regulates mTORC1 [35]. Furthermore, a high percentage of de novo PEComas can have somatic mutations in the TSC1/2 complex. This provided the basis for their initial treatment with rapamycin. 

Rapamycin (sirolimus) was one of the first mTOR inhibitors discovered and was developed initially as an immunosuppressive drug due to its ability to block T-cell activation. It was initially approved by the FDA in 1997 to prevent allograft rejection for use in transplantation before its application as an anti-cancer agent was explored. However, rapamycin is poorly water soluble with limited bioavailability, and the anti-tumor activity in solid tumors is limited, resulting in different analogs of rapamycin, such as temsirolimus and everolimus, to be developed [36]. 

On 22 November 2021, the FDA approved *nab*-Sirolimus for the treatment of locally advanced unresectable or metastatic malignant perivascular epithelioid cell tumor (PEComa) after efficacy was evaluated in the AMPECT (NCT02494570) study (Table 1). *nab*-Sirolimus is a nanoparticle albumin-bound rapamycin particle with a distinct PK profile, significantly higher anti-tumor activity, intratumoral drug accumulation, and mTOR target suppression [37]. The AMPECT study was a phase II, single-arm prospective study to evaluate the efficacy and safety of *nab*-Sirolimus, ultimately leading to its approval. Patients with malignant PEComa were treated with 100 mg/m^2^ *nab*-Sirolimus intravenously once weekly for 2 weeks in 3-week cycles. The investigators observed an overall response rate of 39%, a median PFS of 10.6 months, a median OS of 40.8 months, and no grade ≥ 4 treatment-related events [38].

### 2.4. Chromatin Remodeling

#### 2.4.1. Basic Science

DNA accessibility and regulation are primarily maintained by two classes of enzymes: the switch/sucrose non-fermentable (SWI/SNF) complexes and the polycomb repressor complex 2 (PRC2). Mutations in genes encoding subunits of the SWI/SNF chromatin remodeling complexes have been recognized to play a prominent role in tumorigenesis. SWI/SNF complexes are key regulators of nucleosome positioning and play roles in transcriptional programs such as mediating cell differentiation and lineage specification. A nucleosome is the basic unit of chromatin, consisting of eight histone units in which 147 base-pairs of DNA are wrapped. This highly organized and ordered arrangement of DNA allows for the careful regulation of gene expression. The SWI/SNF complex uses ATP hydrolysis to remodel the chromatin by mobilizing nucleosomes [39]. 

PRC2 is a histone-modifying enzyme composed of three essential core protein subunits: enhancer of zeste 2 (EZH2), embryonic ectoderm development (EED), and suppressor of zeste 12 (SUZ12). EZH2 is responsible for catalyzing the addition of methyl groups onto lysine residues of proteins, EED helps recruit PRC2 to sites within chromatin that need repression, and SUZ12 stabilizes EZH2 through allosteric inhibition [40]. PRC2 is the only known methyltransferase responsible for epigenetic silencing through the catalyzation of methylating histone H3 on lysine 27 (H3K27) on three different levels. 

Inactivation or mutations in genes encoding for SWI/SNF subunits have been identified in multiple human cancers. Biallelic inactivating mutations in *SMARCB1* were identified in malignant rhabdoid tumor (MRT), a highly aggressive pediatric STS [41]. The various subunits of the PRC2 complex have been reported to be overexpressed and dysregulated in multiple types of cancer, including prostate, breast, lung, gastric, bladder, and pancreatic cancer. Studies have shown that epithelioid sarcomas are characterized by genetic aberrations in the *SMARCB1 (INI-1)* gene, which is a potent tumor suppressor gene and part of the SWI/SNF complex. Loss of this gene results in the increased expression and recruitment of EZH2 to key genes to become trimethylated on H3K27 and repressed, thereby upregulating several oncogenic signaling pathways [42,43].

#### 2.4.2. Treatment Applications

##### Tazemetostat for Epithelioid Sarcoma

Epithelioid sarcomas (ES) are malignant tumors with mixed differentiation of both mesenchymal and epithelial cell types, with up to 90% of cases demonstrating a loss of integrase interactor-1 (INI-1) expression. The slow-growing tumor is locally invasive and frequently metastasizes to regional lymph nodes and distant sites, most commonly the lungs. ES are rare and represent less than 1% of STS, are more commonly found in younger males, and are associated with a poor outcome. Loss of INI1 function in ES allows the epigenetic modifier EZH2 to become an oncogenic driver in tumor cells [42].

On 23 January 2020, the FDA granted accelerated approval for tazemetostat (TAZVERIK, Epizyme, Inc., Cambridge, MA, USA) for adults and pediatric patients ≥ 16 years with metastatic or locally advanced ES not eligible for complete resection (Table 1). Tazemetostat is a first-in-class targeted epigenetic regulator that specifically inhibits EZH2 [44]. Data from a phase II open-label trial with locally advanced or metastatic ES treated with tazemetostat (NCT02601950) demonstrated an objective response rate (ORR) of 15%, a disease control rate (DCR) of 26%, and median OS of 82.4 weeks. Despite the modest ORR, based on the impressive DCR in this otherwise chemorefractory tumor, tazemetostat was given accelerated approval for the treatment of advanced/metastatic ES, with a recommended dosing of 800 mg BID [45,46].

### 2.5. Notch Pathway

#### 2.5.1. Basic Science

The Notch signaling pathway was first well established in the study of T-cell acute lymphoblastic leukemia (T-ALL), and its tendency to harbor activating mutations within the *NOTCH1* locus results in the neoplastic transformation of T cells [47]. However, the presence of observed Notch signaling mutations and its role in tumorigenesis in solid tumors have been less studied. The human Notch receptor family consists of four distinct type 1 transmembrane receptors, NOTCH1–4. Notch signaling is initiated by the cell-to-cell contact and interaction between a Notch receptor and 1 of 5 known Notch ligands: jagged 1 (JAG1), JAG2, Delta-like 1 (DLL1), DLL3, and DLL4. Once a Notch ligand binds to its receptor, it undergoes a series of cleavages that results in the release of the active Notch intracellular domain (NICD) to translocate to the nucleus of the cell, thereby initiating a signaling cascade that can interact with other oncogenic pathways, such as those involved with cell cycle regulation, inhibition of apoptosis, and the regulation of angiogenesis, some of which have been discussed earlier [48]. 

The third and last cleavage (S3) of the Notch receptor that initiates its release into the nucleus, thereby activating these pathways, is carried out by the presenilin-γ-secretase complex. It is this S3 cleavage that is the target of a class of compounds known as γ-secretase inhibitors, or GSIs. Targeting the Notch pathway with GSIs has been found to lead to impaired cancer cell growth and tumor progression in multiple tumor models. For instance, GSIs were able to directly induce apoptosis in Kaposi sarcoma both in vitro and in vivo [49]. The role of GSIs in preventing Notch signaling and therefore driving tumor cell differentiation and reducing tumor cell burden and progression has led to increased studies of this pharmaceutical class of compounds in the treatments of various solid tumors, including STS.

#### 2.5.2. Treatment Applications

##### Nirogacestat for Desmoid Tumors

Desmoid tumors (DT), also known as aggressive fibromatosis, are mesenchymal neoplasms with clonal fibroblastic proliferation with infiltrative growth that are predominately locally invasive and do not metastasize. Despite their lack of metastatic potential, desmoid tumors are still quite invasive and can cause significant morbidity and mortality. Desmoid tumors are more common in women compared to men and can occur in both intra-abdominal and extra-abdominal locations. In most asymptomatic patients, watchful waiting can be an appropriate method of management for desmoid tumors [50]. However, first-line treatment for more aggressive desmoid tumors can involve local ablative procedures, radiation, systemic therapy, or in select cases, surgery. Historically, desmoid tumors have not been found to be chemosensitive, with limited responses to doxorubicin [51]. Notch signaling in desmoid tumors have not been thoroughly elucidated; however, a few preliminary studies have shown them to express NOTCH1 and its downstream target HES1 [52].

Furthermore, a phase 1 clinical trial demonstrated a partial response in five of seven desmoid patients with administration of an oral gamma-secretase inhibitor (GSI), nirogacestat (PF-03084014) (Table 1) [53]. There are currently no FDA-approved treatments for the treatment of desmoid tumors, although nirogacestat remains a promising new treatment, having received Orphan Drug Designation from the FDA following promising results from the DeFi randomized controlled phase 3 trial (NCT03785964). Nirogacestat is a novel oral GSI shown to have antitumor activity in patients with desmoid tumor. The DeFi study enrolled a total of 142 patients with progressing desmoid tumor, randomized 1:1 to either nirogacestat or placebo. Nirogacestat demonstrated significant improvement in PFS, ORR, and clinical symptoms compared to placebo [54]. Results from this landmark study were most recently presented at the 2022 European Society for Medical Oncology (ESMO) Congress. Although the median Kaplan–Meier estimate of PFS was not reached in the nirogacestat arm, nirogacestat was able to demonstrate an overall 71% reduction in the risk of disease progression (hazard ratio (HR) = 0.29 (95% CI: 0.15, 0.55); *p* < 0.001) and a confirmed objective response rate of 41% compared to 8% for the placebo. Furthermore, nirogacestat demonstrated significant improvements in key patient-reported outcomes (PRO), most notably improvements in pain, physical/role functioning, and overall health-related quality of life, with 95% of all treatment-emergent adverse events being reported as Grade 1 or 2. Of note, ovarian dysfunction defined by events of amenorrhea, premature menopause, menopause, and ovarian failure was observed in 75% of women of childbearing potential treated with nirogacestat, although these events resolved in 74% of the affected patients, including 100% resolution in those who discontinued the treatment for any reason [55]. 

### 2.6. Immunoncology: PD-1, CTLA-4

#### 2.6.1. Basic Science

In recent years, there has been a significant rise in the interest of developing anti-cancer immunotherapies against STS in response to limited benefits with conventional chemotherapy and radiation. Through the modulation of the body’s immune system against tumor cell growth and survival, the aim is to create an anti-tumor immune response that is more effective and longer lasting. However, tumor cells have developed multiple mechanisms in which they are able to proliferate while evading and inhibiting the body’s natural immune response. Further discerning the mechanisms in which tumor cells can downregulate immune recognition can aid the development of potentially effective immunomodulating therapies against STS.

Cytotoxic T-lymphocyte-associated antigen 4 (CTLA-4) is an immune checkpoint molecule expressed on T_reg_ and conventional activated T-cells. CTLA-4 competes with CD28 in its binding of CD80 and CD86, where it then produces inhibitory signals targeted at activated T-cells. These inhibitory signals force activated T-cells to undergo anergy and apoptosis [56]. Conversely, CD28 binding to CD80 and CD86 produces a costimulatory response responsible for the proliferation of T cells and their activation. However, CTLA-4 has a much stronger affinity than CD28, and once bound to CD80 or CD86, CTLA-4 switches off antigen-presenting cells (APCs) and attenuates T-cell activation. T_reg_ cells in the tumor microenvironment therefore suppress the immune system and the activity of cytotoxic T lymphocytes (CTLs) responsible for killing cancer cells. Furthermore, T_reg_ cells are found to be drastically increased in response to early stages of tumor cell development and progression [57]. 

Another immune checkpoint molecule implicated in the tumorigenic immune response is the surface receptor programmed cell death-1 (PD-1) and its ligand, programmed cell death ligand-1 (PD-L1). PD-1 is a surface protein expressed on already-activated T and B cells, whereas PD-L1 is primarily expressed on antigen presenting cells (APCs) but can also be expressed on tumor cells. PD-1 is a crucial receptor responsible for the maintenance of peripheral tolerance, or the process in which self-reactive T-cells in the periphery are deleted or become anergic [56]. PD-1 and PD-L1 are implicated in a complex process, resulting in immunosuppression against certain tumor types, making it a potential therapeutic avenue for certain histological subtypes of STS. 

#### 2.6.2. Treatment Applications

##### PD1/PD-L1 and CTLA-4 Inhibitors

Nivolumab (trade name Opdivo; Bristol-Myers Squibb) is a genetically engineered anti-PD-1 monoclonal antibody that binds PD-1 with high affinity, thereby blocking its interaction with both PD-L1 and PD-L2 (Table 1). It is currently FDA-approved for multiple indications, including unresectable/metastatic melanoma, non-small cell lung cancer (NSCLC), advanced renal cell carcinoma (RCC), and numerous others [58]. Pembrolizumab (trade name Keytruda; Merck) is a humanized monocolonal IgG4 kappa antibody that also targets PD-1 (Table 1). Pembrolizumab is currently FDA approved for similar indications, including melanoma, renal cell carcinoma, and urothelial carcinoma [59]. Ipilimumab (trade name Yervoy; Bristol-Meyers Squibb) is a CTLA-4 antibody that prevents CD80 and CD86 on APCs from binding to CTLA-4 on T cells (Table 1). This results in T-cell activation and proliferation, as well as amplification of T-cell-mediated immunity. It is currently FDA approved for multiple indications, including melanoma, RCC, and others [60]. 

Immunotherapy given alone as monotherapy has shown limited efficacy in an unselected population of STS; however, the ability to combine multiple immunotherapies together, or with other cytotoxic treatments, has shown greater promise. The ALLIANCE A091401 (NCT02500797) was an open-label, multicenter, randomized phase II study in the US investigating the efficacy of nivolumab with or without ipilimumab in unresectable or metastatic STS across various histological subtypes. The results demonstrated radiological response in 6/38 (16%) of patients treated with combination nivolumab-ipilimumab, surpassing the prespecified primary endpoint ORR of 13%. The number of confirmed responses in the nivolumab monotherapy group was 2/38 (5%) patients. The authors of the study concluded that further investigations into nivolumab monotherapy was unwarranted in an unselected group of STS. Of note, of the eight responses seen across the two groups, three were in patients with undifferentiated pleomorphic sarcoma and three were in patients with leiomyosarcoma, suggesting that this combinatorial therapy may be more active in certain STS subtypes [61,62]. 

Interestingly, pembrolizumab monotherapy has shown modest efficacy in STS subtypes in the multicenter, single-arm phase II trial SARC028 (NCT02301039). The STS cohort in the study demonstrated an ORR of 18%, varying across multiple histological subtypes—most notably 40% ORR in undifferentiated pleomorphic sarcomas (1CR, 3PR), and 20% in liposarcomas (2PR). The authors of the study concluded that pembrolizumab demonstrated meaningful clinical activity in both undifferentiated pleomorphic sarcoma and liposarcoma subtypes, which warrants further studies in expansion cohorts of these subtypes to better characterize their efficacy [63].

### 2.7. CSF-1 Abberations

#### 2.7.1. Basic Science

Colony stimulating factor-1 (CSF-1) is produced by certain cancers and is responsible for recruiting myeloid cells that suppress antitumor immunity. CSF-1 signaling through binding of its receptor CSF-1R drives pathways involved in cell survival, differentiation, and proliferation. CSF-1 is also responsible for recruiting myeloid-derived suppressor cells (MDSCs) to the tumor microenvironment, where it can then drive tumor progression through suppression of antitumor immune responses and promotion of angiogenesis. Additionally, translocation of the CSF-1 gene has been shown to result in the hyperexpression of macrophage CSF-1, which is implicated in the development of tenosynovial giant cell tumor (TGCT). The overexpression and excessive secretion of CSF1 attracts monocytes and macrophages to the tumor site, a process known as the “landscape effect.” The increased recruitment of inflammatory cells to the tumor site creates the neoplastic landscape that sets the stage for subsequent TGCT development [64]. 

#### 2.7.2. Treatment Applications

##### Pexidartinib for Tenosynovial Giant Cell Tumor

Tenosynovial giant cell tumor (TGCT), formerly pigmented villonodular synovitis (PVNS), is a rare group of benign tumors arising from joint synovia, bursae, and tendon sheaths. Localized forms are most frequently found in the hand, while diffuse forms are commonly found in the knee. Progression of the tumor is typically slow, although it can result in joint deterioration and osteoarthritis. Primary treatment usually involves surgical resection of the pathological tissue, although recurrence rates, especially for diffuse type TGCT, can be relatively high. There are also anecdotal cases where TGCT can undergo malignant transformation and metastasize on rare occasion [65]. TGCT is predominantly caused by a genetic translocation in the stromal cells of the synovial membrane, resulting in the overexpression of colony stimulating factor 1 (CSF-1) that then recruits CSF1R-expressing cells to the tumor mass. This results in clinical manifestations of pain, swelling, limited range of motion, and joint instability [66]. 

Pexidartinib is an orally administered small-molecule tyrosine kinase inhibitor (TKI) that selectively inhibits CSF1R inhibitor and c-kit receptor tyroise kinase (KIT). The randomized phase 3 ENLIVEN study treated 120 symptomatic, advanced TGCT patients for whom surgery was not recommended, to either the pexidartinib or placebo arm (1:1). The overall response rate was higher for pexidartinib (39%) compared to placebo (0%) at week 25, with the best overall response rate of 53% at the 22-month median follow up. The pexidartinib group also reported significant improvements in physical functioning, relative ROM, and stiffness compared to the placebo group. The ENLIVEN study demonstrated pexidartinib as the first systemic therapy to show a robust response in TGCT, with improved patient symptoms. Pexidartinib was subsequently approved by the FDA for adult patients with symptomatic TGCT associated with severe morbidity or functional limitations who are not amenable to improvement with surgery [67]. Other notable therapeutic agents that also target the CSF-1 pathway in TGCT include imatinib, nilotinib, and vimseltinib, although pexidartinib remains the only treatment for TGCT currently approved by the FDA [68].

### 2.8. Genetic Fusions and Alterations

#### 2.8.1. Basic Science

Various histological subtypes of STS have been associated with specific genetic fusions, such as *SS18::SSX* in synovial sarcoma, *FUS::DDIT3* in myxoid liposarcoma, and *TMP4::ALK* in inflammatory myofibroblastic tumor [69]. The advent of NTRK-inhibitors and their superior efficacy regardless of tumor type has highlighted the growing importance of utilizing various molecular testing techniques in STS for *NTRK* fusions as well as other genetic aberrations. Tropomyosin receptor kinase (TRK) is a tyrosine kinase predominantly found in neuronal tissue, with three receptors (TRK A/B/C) encoded by their respective neurotrophic tyrosine receptor kinase genes (*NTRK*) *NTRK 1/2/3* [70]. Mutations or rearrangements in the TRK family have been described as a mechanism of oncogenesis. The product of TRK fusions is a constitutively activated TRK protein that is ligand-independent and has effects on several downstream oncogenic pathways [71]. Although *NTRK* fusion is present in more than 90% of cases of infantile fibrosarcoma, it is much more rare in other adult and pediatric sarcomas, with a frequency of less than <1% [70,72]. For that reason, integrating *NTRK* testing into the management of STS patients has presented diagnostic challenges in the identification of this rare biomarker. 

The *ALK* gene is another gene that encodes a transmembrane tyrosine kinase receptor. Anaplastic lymphoma kinase (ALK) has recently been best described in the context of non-small cell lung cancer (NSCLC). Indeed, numerous ALK inhibitors have been developed for the treatment of metastatic NSCLC and have gained FDA approval for first-line treatment of ALK-positive NSCLC within the last decade [73]. Unfortunately, the normal physiological function of ALK in humans is poorly understood. However, *ALK* gene rearrangements have been described in 50% of inflammatory myofibroblastic tumors, a mesenchymal neoplasm, and therefore relevant to the subject of this review [74]. 

#### 2.8.2. Treatment Applications

##### Crizotinib for ALK-Rearranged Inflammatory Myofibroblasitc Tumors

Inflammatory myofibroblastic tumors (IMTs) are a relatively unknown mesenchymal neoplasm characterized by a spindle-cell proliferation with an inflammatory infiltrate. Roughly half of all diagnosed IMTs carry rearrangements of the anaplastic lymphoma kinase (ALK) locus on chromosome 2p23, resulting in aberrant cellular ALK expression [75]. Crizotonib is a small molecule tyrosine kinase inhibitor targeting ALK and is currently approved for adult patients with *ALK*- or *ROS1*-positive, metastatic non-small cell lung cancer (NSCLC). Crizotinib competitively inhibits adenosine triphosphate from binding to ALK receptors. The European Organization for Research and Treatment of Cancer (EORTC) conducted a prospective phase II study (EORTC 90101 CREATE) that included a subcohort of *ALK*-positive IMT patients treated with crizotinib. At median follow-up for 50 months, the median PFS was 18.0 months and the 3-year overall survival rate was 83.3% [76]. A multicenter, single-arm trial previously evaluated the long-term effects of crizotinib in *ALK*-positive IMTs and reported results of an ORR of 67% with 1 CR and 5 PRs (n = 9), and a median duration of treatment of nearly 3 years [77]. On 14 July 2022, the FDA approved crizotinib for adult and pediatric patients 1 year of age and older for unresectable, recurrent, or refractory inflammatory anaplastic lymphoma kinase (ALK)-positive myofibroblastic tumors (IMTs). 

##### Larotrectinib for TRK-Fusion Positive Cancers

Larotrectinib is a highly selective and potent TRK inhibitor that works by competitively binding to the ATP-binding site of TRKA/B/C. Larotrectinib was FDA approved on 26 November 2018 for the treatment of adult and pediatric patients with solid tumors with *NTRK* gene fusions.

The approval was based on a primary analysis of a set of 55 enrolled adult and pediatric patients with TRK fusion-positive solid tumors across three clinical studies (NCT02122913, NCT02637687, NCT02576431). According to independent review, ORR was 75%, with 71% of the responses ongoing at the 1-year mark. Interestingly, responses were observed across all tumor types regardless of patient age and TRK fusion characteristics. Of note, the study population included 17 unique cancer diagnoses, including infantile fibrosarcoma, gastrointestinal sarcoma, peripheral-nerve sheath tumor, and other soft tissue sarcoma subtypes. This landmark study validated TRK fusions as therapeutic targets and demonstrated that larotrectinb was capable of inducing a durable and effective response with minimal treatment-related adverse events [78].

## 3. Future Directions

### 3.1. Oncolytic Viruses

The emergence of oncolytic viruses to combat multiple mechanisms of diverse cancer types has been a promising new direction for cancer therapy. Oncolytic viruses are organisms that have the ability to invade and lyse tumor cells while still eliciting an immune response. The success of oncolytic viruses was first demonstrated in the use of a genetically modified herpes virus (HSV-1), Talimogene Laherparepvec (T-VEC), which was the first oncolytic virus approved by the FDA for melanoma cancer therapy [79]. T-VEC is a modified HSV-1 virus that has been inserted with two copies of the human granulocyte-macrophage colony-stimulating factor (GM-CSF). The GM-CSF engineered into T-VEC is responsible for enhancing T-cell priming by dendritic cells, resulting in a more robust immune system response against infected cancer cells [80]. Intratumoral injection of T-VEC is currently in multiple clinical trials for the treatment of soft tissue sarcomas in combination with other therapies such as immune checkpoint inhibitors. A phase 2 study of T-VEC in combination with PD-1 inhibitor nivolumab and chemotherapeutic agent trabectedin enrolled 36 evaluable STS patients and observed a median PFS of 5.5 months and a disease control rate of 86.1%, suggesting that combinatorial therapy of T-VEC with immune checkpoint inhibitors and/or chemotherapy can result in better outcomes [81]. Other oncolytic viruses such as adenovirus-based oncolytic viruses have been in clinical development for cancer, although T-VEC remains the most studied in the treatment for STS.

### 3.2. Targeting Fusion Oncoproteins and Oncogenes

The advancements in Next-Generation Sequencing (NGS) and their ability to allow greater accumulation of cancer genetic data has led to the discovery of several different aberrant genetic fusions responsible for tumorigenesis. Fusion oncoproteins involving ALK or NTRK have been incredibly important in their discovery, albeit very rare to find in STS. By further elucidating more of these pathogenic drivers, efforts to develop targeted therapies against them provides a promising avenue towards treatments directly targeting these fusion oncoproteins. Furthermore, by continuing to validate NGS results with certain pathogenic pathway activations, fusions of certain genetic oncogenes can be directly targeted as well. Emerging studies have demonstrated that genome-editing systems, such as clustered regularly interspaced short palindrome repeats (CRISPR) associated protein 9, or CRISPR-Cas9, may have therapeutic potential through direct targeting of fusion oncogenes [12]. CRISPR-Cas9 has therefore emerged as a highly regarded research hotspot in genome editing. It uses a guide RNA (gRNA) to recognize and target specific sections of genomic DNA, where it then recruits the Cas9 nuclease to create double-stranded breaks at the specific DNA location. This relatively simple and programmable method has allowed researchers to study certain disease conditions in vivo through genomic silencing and knock-down/in experiments [82]. However, their utility in vivo remains to be studied, especially in targeting genetic oncofusions of various STS subtypes. Certain subtypes are strongly associated with genetic fusions, such as Ewing’s sarcoma (*EWSR1::FLI1)*, alveolar rhabdomyosarcoma (*PAX3/7::FOXO1)*, and synovial sarcoma *(SS18::SSX1/2/4)*. Preclinical studies have shown some promise utilizing the CRISPR-Cas9 system, such as the subtotal tumor clearance of subcutaneous Ewing’s sarcoma xenografts in mice treated the CRISPR-Cas9 strategy against *EWSR1::FLI1,* demonstrated by Mitra et al. [83]. However, their utility and efficacy in vivo remain to be established.

Although clinical applications of the CRISPR-Cas9 strategy in targeting the *EWSR1::FLI1* fusion in Ewing’s sarcoma remain to be seen, there has been recent development of small molecule inhibitors to target the genetic fusion proteins of *EWSR1::FLI1*. TK216 is a first-in-class small molecular inhibitor that was developed to specifically target EWSR1-FLI1. It is a derivative of YK-4-279, a small molecule that was previously identified to be able to disrupt the interaction between EWSR1-FLI1 and RNA helicase A, thereby inducing apoptotic cell death in Ewing’s sarcoma cell lines [84]. TK216 was recently evaluated in a phase I/II study in which TK216 was administered as monotherapy or in combination with vincristine in patients with relapsed or refractory Ewing’s sarcoma (NCT02657005). After dose escalation, a total of 35 patients were treated at the recommend phase II dose (RP2D) with the optional addition of vincristine. Results demonstrated a DCR of 46.4% and a SD median duration of 113 days. From these results, the investigators concluded that TK216 plus vincristine was well tolerated and showed promising anti-tumor activity in this heavily pre-treated Ewing sarcoma cohort [85].

## 4. Conclusions

The rarity and heterogeneity of STS have complicated the clinical approach to managing this complex disease. The various subtypes of STS pose a great challenge due to their histologically specific signaling pathways, prognostic factors, and inherent variability in responses to conventional treatments. Despite these challenges, advancement in understanding the molecular pathways involved in sarcomagenesis, such as angiogenesis and cell cycle regulation, has allowed for the development of far more effective, targeted approaches to various histological subtypes of STS. The advancement of molecularly targeted therapies has changed the way clinicians approach the treatment of this rare disease, with a decreasing emphasis on broad cytotoxic chemotherapies. NGS and other molecular diagnostic technology has allowed pathologists to further differentiate between sarcoma subtypes and identify specific key genetic drivers, allowing for clinicians to take a more personalized medicine approach aimed at tailoring the medical treatment to the individual characteristics of the disease biology. With the ongoing development of future therapeutics and technologies aimed at molecular targets, combinatorial treatments involving targeting multiple pathways in combination with established chemo- or immunotherapies further provides a promising multifaceted approach to managing this complex malignancy.

## Figures and Tables

**Table 1 cancers-15-01692-t001:** Molecular pathways and associated therapies in STS.

Molecular Pathway	Targeted Treatments
Cell Cycle Regulation	Palbociclib, Abemaciclib, Milademetan
Angiogenesis	Pazopanib, Regorafenib, Cabozantinib, Sorafenib
Growth Factors	*nab*-Sirolimus
Chromatin Remodeling	Tazemetostat
NOTCH	Nirogacestat
Immunoncology (PD-1/CTLA-4)	Nivolumab, Pembrolizumab, Ipilimumab
CSF-1	Pexidartinib
Genetic Fusion/Alteration	Crizotinib, Larotrectinib

## Data Availability

Data sharing not applicable.

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
