# Peer review of "The Next Frontier in Sarcoma: Molecular Pathways and Associated Targeted Therapies"

_cancers, 2023, doi:10.3390/cancers15061692_

Round 1

Reviewer 1 Report

Thank you for submission of your manuscript. It is extremely well written and addresses various interesting topics in soft tissue sarcomas. This paper shows that further development of molecularly targeted therapies could change the clinician's approach to the treatment of sarcoma. This review will provide comprehensive and useful information to our readers. It is well written besides that the references are too brief. The other thing that would be better to rewrite is that the first choice of treatment for desmoid is active surveillance (watchful waiting), not surgery, so as not to mislead the readers.

Author Response

Thank you for the review, appreciate the comments. We have added a line to the section "Nirogacestat for Desmoid Tumors" stating:

In most asymptomatic patients, watchful waiting can be an appropriate method of management for desmoid tumors (Kasper et al., 2011). However, first-line treatment for more aggressive desmoid tumors can involve local ablative procedures, radiation, systemic therapy, or in select cases, surgery.

Reviewer 2 Report

The authors have performed an interesting review of the state of the art of molecular targeted therapies for soft tissue sarcomas. They discuss the various molecular pathways involved in the sarcomagenesis and treatment applications. They emphasize the importance of a well understanding of the molecular biology of sarcoma in order to develop relevant targeted treatments. Overall, even if this the review is comprehensive and well-written, it would be of interest to add a table to give the reader a visual summary of the molecular pathways and related targeted treatments. Furthermore, I think that further aspects need to be revised:

1.      Line 9 : in addition of the statement “over than 75 different histological subtypes”, it would be important to precise that there is more than 150 molecular subtypes according to the last WHO classification

2.      Line 26 : please precise “FNCLCC” grade

3.      Line 30 : please precise “en-bloc R0 resection”

4.      Line 31 : adjuvant radiotherapy is indicated for grade 2 and grade 3

5.      Line 35 : according to the latest phase III trial (doi: 10.1001/jama.2020.1707), OS = 20 months

6.      Cell cycle regulation: could the authors also discuss about MDM2 inhibitors in liposarcoma ? (DOI: 10.1007/s12094-019-02158-z; Journal of Clinical Oncology 2022 40:16_suppl, 3004-3004)

7.      Angiogenesis : the authors focused on STS and Pazopanib (PALETTE) but other significant studies with antiangiogenics are important : REGORAFENIB (DOI:https://doi.org/10.1016/S1470-2045(18)30742-3) and CABZOANTINIB (doi: 10.1016/S1470-2045(19)30825-3) for bone sarcomas, SORAFENIB (DOI: 10.1056/NEJMoa1805052) and PAZOPANIB (doi.org/10.1016/S1470-2045(19)30276-1) for desmoid tumors

8.      CSF1-aberrations : other teraghted treatemnts such as Imatinib (doi.org/10.1038/s41598-019-51211-y), Nilotinib (doi.org/10.1016/S1470-2045(18)30143-8) and Vimseltinib (Annals of Oncology (2022) 33 (suppl_7): S681-S700. 10.1016/annonc/annonc1073) should be cited

9.      Gene fusions and alterations : Following rules of gene fusion nomenclature names, authors should use double colons (::)

Author Response

Thank you for the review. We have added a Table 1 with a summary of the Molecular Pathways and matching Targeted Treatments.

  1. This line has been edited to change to 150 subtypes
  2. Edited to add FNCLCC
  3. Edited to add en-bloc R0 resection
  4. Added adjuvant radiotherapy
  5. Added ANNOUNCE data at 20 months.
  6. Added section on Milademetan with data on published Phase 1 study and note of ongoing Phase 3 study.
  7. Added section on other anti-angiogenic agents such as regorafenib, cabozantinib, and sorafenib
  8. Added sentence on other CSF-1 inhibitors such as imatinib, nilotinib, and vimseltinib. However, since pexidartinib is the only FDA approved agents, only put pexidartinib in the table.
  9. All gene fusions changed to double colon
